# Construction and Ion Transport-Related Applications of the Hydrogel-Based Membrane with 3D Nanochannels

**DOI:** 10.3390/polym14194037

**Published:** 2022-09-27

**Authors:** Yushuang Hou, Shuhui Ma, Jinlin Hao, Cuncai Lin, Jiawei Zhao, Xin Sui

**Affiliations:** College of Materials Science and Engineering, Qingdao University, Qingdao 266071, China

**Keywords:** hydrogels, 3D structure, ion channels

## Abstract

Hydrogel is a type of crosslinked three-dimensional polymer network structure gel. It can swell and hold a large amount of water but does not dissolve. It is an excellent membrane material for ion transportation. As transport channels, the chemical structure of hydrogel can be regulated by molecular design, and its three-dimensional structure can be controlled according to the degree of crosslinking. In this review, our prime focus has been on ion transport-related applications based on hydrogel materials. We have briefly elaborated the origin and source of hydrogel materials and summarized the crosslinking mechanisms involved in matrix network construction and the different spatial network structures. Hydrogel structure and the remarkable performance features such as microporosity, ion carrying capability, water holding capacity, and responsiveness to stimuli such as pH, light, temperature, electricity, and magnetic field are discussed. Moreover, emphasis has been made on the application of hydrogels in water purification, energy storage, sensing, and salinity gradient energy conversion. Finally, the prospects and challenges related to hydrogel fabrication and applications are summarized.

## 1. Introduction

Hydrogel is a flexible substance with high water retention and exists in the form of a three-dimensional (3D) crosslinked polymer network. It maintains its structural stability without dissolving when loaded with a certain amount of water [1]. Thus, hydrogels show properties similar to both liquid and solid [2,3]. The crosslinked polymer chain structure gives the hydrogel elastomer properties and provides mechanical support in 3D channels, while water retention ensures the permeability of chemicals and ions in a hydrogel [4]. Initially, the distinction between hydrogels and polymers was vague, and the two concepts were often used interchangeably [5]. Later, with more in-depth research, the understanding of hydrogel gradually became clear and mature. Poly (hydroxyethyl methacrylate) (PHEMA) hydrogels were synthesized first for biomedical applications by Wichterle and Lim in 1960 [6]. The discovery of the extracellular matrix (ECM) as a natural material resembling hydrogels in living organisms has led to the realization that the origin of hydrogels predates their definition, and the hydrogel is a substance that may be derived from organisms [7,8].

At present, biocompatible and environmentally friendly natural hydrogel materials such as sodium alginate [9], chitosan [10], cellulose [11], and protein [12] have been discovered [13]. Some synthetic hydrogels, such as polyvinyl alcohol [14] and polyacrylamide [15], were designed to overcome the disadvantages of natural materials or to obtain specific components, to broaden their application. For example, synthetic hydrogels are more stable than natural hydrogels and can be designed to incorporate certain functional groups that are nonexistent in natural materials [16]. The molecular structure design of hydrogels has always been a point of interest. The formation of a networked structure of hydrogels involves a polymerization between small molecules and crosslinking between polymers, which is related to different initiation conditions such as initiator, illumination, etc., and crosslinking mechanism (physical or chemical crosslinking) [17]. The type of polymer chain determines whether the network configuration of hydrogel is single, double, or even interpenetrating [18]. The distribution and morphology of the pore size are also controlled by the preparation conditions [19]. 

Hydrogels are tunable at the molecular level. Their unique 3D structure and designable functional groups are conducive to their applications in ion transport. The large specific surface area provided by the 3D network structure endows hydrogels with adsorption capability [20]. In the treatment of dyes and heavy metal ions, the modification of groups and the selection of polymer molecular structure can achieve the selective removal of multiple dyes and ions as well as some specific substances [21,22]. Jiang et al. developed hydrogels as ion-selective membranes to demonstrate salinity gradient energy conversion with the help of RED technology to obtain clean and renewable energy from seawater [23]. Hydrogels are also used as an electrolyte for typical energy storage devices such as batteries and supercapacitors [24,25]. High ionic conduction of hydrogel electrolytes is achieved from its unique 3D interconnection structure and channels for the ion movement provided by high water content [26]. In addition, by incorporating special materials or introducing specific functional groups, hydrogels can be endowed with intelligent environmental response characteristics (pH, temperature, light, electricity, magnetism, etc.) [27,28]. This review briefly describes the sources of hydrogel materials, the internal mechanisms involved in the synthesis of hydrogels, the spatial network structure, and the characteristics of hydrogels. In addition, applications such as water treatment, salinity gradient energy conversion, energy storage, and sensors are highlighted. Hydrogel-based materials are expected to come out as outstanding candidate materials to replace traditional rigid devices in the next generation. 

## 2. Source of Hydrogel Material

The concept of hydrogel was first proposed in 1894, but its origins date back to an earlier period [29]. In living organisms, a complex network around cells is called the extracellular matrix (ECM). As the external environment for the survival of cells, it not only binds tissues or organs together with cells as a simple support scaffold but also contains many signal molecules, which transmit information through the ECM to regulate cell growth, metabolism, and other activities. The main structure of the ECM comprises a fibrin network composed of fibronectin, collagen, fibrin, laminin, and elastin [30]. This nanoscale network structure enables signaling molecules to be transmitted in a cell-stroma-cell pathway. The ECM is a typical natural hydrogel matrix. In addition, there are many natural hydrogels, such as chitosan, cellulose, alginate, hyaluronic acid, some natural hydrophilic polymer polypeptides, etc. [31]. The natural materials mentioned above have some common advantages. They are abundant in nature, economically available, environmentally friendly, and biodegradable [32]. Especially as biomedical materials, hydrogels based on natural materials are highly compatible with organisms [33]. However, they are limited by poor stability and mechanical strength with a limited variety and number of functional groups [34]. Some applications, such as cell culture, require the medium to be biologically inert and therefore cannot use a naturally sourced biological substrate [35]. In such cases, hydrogel materials have been artificially synthesized with specific functional groups for different purposes. Although synthetic hydrogels are not as biocompatible as natural materials, their tunable structures and properties could be developed on demand. Synthetic polymer materials are rich in variety, such as poly-alcohols (polyvinyl alcohol, polyethylene glycol), polyacrylic acid (PAA), and its derivatives, polyacrylamide (PAM), poly (hydroxyethyl) methacrylate (PHEMA), and so on [36]. 

Nowadays, research is not only limited to developing single-source hydrogels but to designing hybrid hydrogels by combining two materials from different sources in a certain way. This hybrid strategy demonstrates the double modification of two separate materials to achieve complementary advantages. Zhang et al. implemented the agar-C_3_N_4_ hybrid hydrogels strategy for pollution treatment to achieve a hydrogel network with pollutant adsorption and degradation capabilities [37]. The hybridization strategy is relatively simple: the 3D crosslinked hydrogel is synthesized by heating-cooling polymerization of agar and graphic carbon nitride g-C_3_N_4_ nanoparticles. Pure natural agar hydrogels modified with photocatalyst g-C_3_N_4_ have the advantages of high adsorption capacity and good photodegradation ability. Compared with the pure g-C_3_N_4_, the degradation rate of methylene blue (MB) by hybrid gel increased by 4.5 times. This can be attributed to the synergistic effect of adsorption and photocatalytic degradation. Various integration strategies of natural and synthetic materials render hydrogels more widely utilizable due to the compatibility of properties [38]. 

## 3. Construction of Hydrogel Matrix Module 

### 3.1. Crosslinking Mechanism 

The construction of a hydrogel network involves a series of crosslinking reactions. From the starting material of crosslinking reaction, it could be divided into monomer polymerization crosslinking and prefabricated polymer chain crosslinking [39,40]. The hydrogel polymers formed by different monomers are also different. For instance, homopolymer hydrogel is formed with a single monomer composition type, while copolymer hydrogel is formed with two or more monomers [41,42]. The crosslinking between polymer chains can be divided into physical and chemical crosslinking according to whether or not there is a covalent interaction [43]. The gel formed by the covalent bonding of the polymer chain is called chemical hydrogel. This kind of bond will not be destroyed easily, so the structure of the hydrogel is relatively stable and shows higher mechanical strength, also known as permanent hydrogel. Unlike physical hydrogels, chemical reactions are non-spontaneous, and covalent bonds are formed by reactions between functional groups, which require certain conditions to initiate. Initiation conditions leading to the formation of a chemical crosslinking network are divided into physical conditions (radiation or illumination) (Figure 1a) and chemical conditions (initiator and crosslinking agent) [44]. Under high energy radiation, active centers are formed to initiate the interaction of polymer chains, forming crosslinking networks. High-energy electron beams and gamma rays are the main radiation conditions to induce the cross-linking of the polymer network. Under irradiation conditions, the water particles in the gel solution produce living groups, which react with the polymer by hydrogen extraction to form polymer free radicals, and the free radicals are cross-linked to form hydrogel network polymers. Crosslinkers are small molecular compounds, such as formaldehyde, glutaraldehyde, etc., capable of forming covalent bonds with polymer chains. Physical crosslinking is also called noncovalent crosslinking. The crosslinking between polymers relies on hydrogen bonding, van der Waals forces, electrostatic attraction, and hydrophobic interaction. The product is also known as a reversible gel because the interchain interaction can be affected by temperature and environment pH value. Physically crosslinked hydrogels also have their unique advantages. No crosslinking agent or other chemical substance is needed. Hence, the product is relatively safe and can be used in biological engineering/tissue engineering. The freeze–thaw technique is a common method for forming a physical crosslinking network [45]. By freezing polyvinyl alcohol (PVA) aqueous solution, the ice in the amorphous region outside the crystallization zone forces crystallization between the polymer chains (Figure 1b) [46]. These crystallization points are the physical crosslinking points, which will not be destroyed after thawing to form PVA hydrogel. To improve the quality of physical crosslinking, the hydrogel is fabricated via multiple freeze–thaw cycles. Mastrangelo et al. demonstrated that three freeze–thaw cycles were needed to obtain thick walls and large pores due to the increased crosslinking and phase separation (Figure 1c) [47].

### 3.2. Spatial Network Structure 

When the crosslinked framework of a hydrogel consists of only one kind of polymer chain, it is called a single network (SN) hydrogel. In contrast, double network (DN) hydrogels are constructed from two polymer networks with distinct structures and properties [48]. In general, the first network of DN hydrogels is a rigid polyelectrolyte that acts as a scaffold. The second network is a functional, flexible polymer (Figure 1d) [49]. In comparison to the simple single polymer network structure, the dual network is improved in both mechanical strength and toughness due to the presence of the second network. At present, it is applied in biomedicine, biomimetic machinery, and other fields, which have a higher demand for strength and toughness. Li et al. assembled conductive poly(3,4-ethylenedioxythiophene): polystyrene sulfonate (PEDOT: PSS) and poly (vinyl alcohol) (PVA) to obtain PEDOT: PSS/PVA DN hydrogel, achieving high electrical conductivity with a large fracture strain [50]. DN hydrogel was comprehensively compared with highly conductive SN hydrogel and the conductive base interpenetrating network polymers (IPN) hydrogel network. It exhibits both high conductivity and high tensile properties. IPN structure is very similar to the double network structure. It is composed of two or more polymer chains. There are usually two kinds of synthesis methods. One is to mix the prepolymer component monomer, initiator, and crosslinking agent for in situ polymerization and crosslinking. The other method polymerizes different polymer components first, followed by crosslinking [51,52]. The molecular chains of different polymers in IPNS are “independent” and “close”. Different polymers have distinctive phases and do not chemically combine, but they interlock and combine into a 3D network, which cannot be separated independently. The kind of physical structure formed is characterized by the unique forcing effect of the topological structure, which can form stable binding between two or more polymers with very different properties or different functions, thus achieving special coordination in properties and structures, and greatly improving mechanical strength. In addition, if there are linear molecules in the polymer network structure, it is called a semi-interpenetrating network (semi-IPN) [53]. 

**Figure 1 polymers-14-04037-f001:**
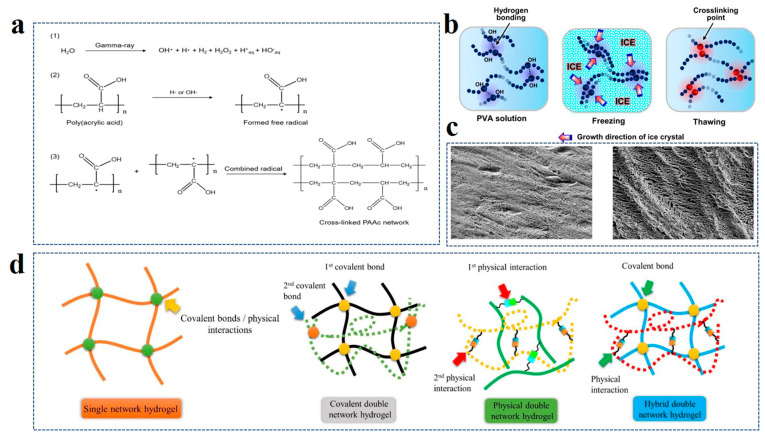
(**a**) Synthesis routes of chemical cross-linked PAAc hydrogel networks by ray initiation. Reprinted with permission from Ref. [44]. (**b**) Mechanism of physical cross-linked hydrogel formation in PVA aqueous solution by the freezing-thawing method. Reprinted with permission from Ref. [46]. (**c**) Scanning electron microscopy (SEM) of hydrogel formed by single freezing-thawing (left) and three freezing-thawing cycles (right). Reprinted with permission from Ref. [47]. (**d**) Comparison of polymer chain composition of a single network and double network hydrogels. Reprinted with permission from Ref. [49].

## 4. Characteristics and Properties

### 4.1. Water Content

Hydrogels are polymers network dispersed within the water. Water molecules are wrapped in a three-dimensional network of entangled polymer chains [54]. It has a strong water-retention capacity of over 90%. In arid areas, hydrogels can be used as an efficient water-retaining agent [55]. The water absorbed by their structure can be released slowly into the environment, achieving the effect of drought resistance. In addition, the hydrogel also showed good flame-retardant performance. Vlassak et al. laminated polyacrylamide-alginate hybrid hydrogels on fabrics to create flame-retardant wearables [56]. The water contained in the hydrogel layer has a large heat capacity and enthalpy of evaporation. When it encounters an open flame, the water on the outer surface of the hydrogel evaporates and takes away a lot of heat (Figure 2a), while the fabric layer with low thermal conductivity blocks the inner hydrogel from transmitting heat inwards, which prevents the human body from being burned at high temperature.

Many properties of hydrogels are related to their water content. Zhou et al. explored the relationship between water content and adhesion by regulating the water content of PAM hydrogel [57]. Their work demonstrated that the relationship between adhesion energy and moisture content was not simple. It is affected by the synergistic influence of four factors controlled by water content, such as surface chain density, volume energy dissipation of hydrogel, and the effective contact area at the interface between hydrogel and matrix (Figure 2b). When the water content of the gel increases, the polymer chains expand, the surface chain density decreases, and the adhesion decreases. The effect of high water content on adhesion between the hydrogel and solid matrix is similar to polymeric liquid films. The corresponding adhesion is small when the total number of water molecules is small. Increasing water content increases the volume energy dissipation, and the adhesion force decreases; the hydrogel with low water content has a rough surface, incomplete contact with the rigid medium, few binding sites, and low adhesion. 

Ueda et al. investigated the influence of water content on the mechanical properties of supramolecular hydrogels (Figure 2c) [58]. At lower water content, the gel shrinks, the polymer chains bind tightly, and cannot move freely, nor can they interact with the host and guest. Energy dissipation only occurs when the covalent bond is broken, and the hydrogel exhibits low toughness. The hydrogel expands excessively with reduced toughness when the water content is high. With the increase in water absorption, the free volume of water in the polymer chain increases [59]. The increased free body water acts as a plasticizer, making the polymer molecular chains slip so that the molecular chain movement is enhanced [60,61]. Only host–guest interaction can occur in this case. The moderate water content provides the right space for the free chain to move and for the host–guest interaction to dissipate energy. Therefore, the water content that is too low or too high decreases the mechanical properties. High mechanical toughness can only be achieved with appropriate water content. 

### 4.2. 3D Porous Structure

Hydrogel has a loose network of porous structures, and these microscopic pores are interconnected in 3D space. pore size is affected by many factors. During the preparation process, polymer concentration will affect the pore size of a hydrogel [62]. The solution with higher polymer concentration has a smaller average pore size because fewer free water phases can crystallize in a high concentration solution. In addition, low temperature is conducive to forming denser and smaller pores due to the solvent’s faster crystallization rate at low temperatures with more uniform nucleation. At the same time, the size and orientation of the pore are also affected by refrigeration technology. When preparing porous hydrogels, the hydrogel pores obtained by the conventional copolymerization method are randomly distributed, and the size is irregular. The porous structure with controllable size and regular distribution can be obtained by directional freezing technology by controlling the time and location of ice crystal growth (Figure 3a) [63]. Zhang et al. obtained a well-arranged porous PVA hydrogel combination (Figure 3b) by limiting the growth direction of ice crystals by directional freezing and demonstrated the relationship between the freezing rate of hydrogel and channel spacing [64]. It can be seen in Figure 3c, that the pore size of hydrogel decreases with the increase in freezing rate. This is due to the small but uniform crystallization caused by rapid cooling.

### 4.3. Ion Charge Carrier

Hydrogels can carry electrical charges through functional groups and can be used for ion transport applications [65]. Sodium alginate is a natural linear polysaccharide molecule that exhibits polyanionic behavior in an aqueous solution due to a large amount of -COO^−^ and can be rapidly crosslinked by Ca^2+^ to form a gel structure [66]. Chitosan, containing free amino groups is cationic and is the only natural cationic polymer derived from animals. It can be chemically modified for wider applications [67]. Non-ionic hydrogels, also known as neutral hydrogels, are comprised of neutral polymers, such as polyvinyl alcohol (PVA), polyethylene glycol (PEG), polyethylene oxide (PEO), polyhydroxyethyl methacrylate (PHEMA), N,N-dimethylacrylamide (DMAAm), etc. [68]. 

Zwitterionic ionic hydrogel has a polymer chain structure containing cationic and anionic groups simultaneously [69]. Due to the coexistence of cation and anion, the number of positive and negative charges is equal, and is overall electrically neutral, bearing the advantages of both ionic and neutral polymers. Hydrogels of zwitterionic polymers show higher hydrophilicity than other kinds of ionic hydrogels [70]. However, due to the tendency of self-association between opposite charges, interchain and in-chain polymers may attract each other and curl, resulting in densification, which is unfavorable in some applications. Recently, Aleid et al. developed zwitterionic poly-[2-(methacryloyloxy)ethyl]-dimethyl-(3-sulfopropyl)ammonium hydroxide (PDMAPS) for atmospheric water collection [71]. It contains the zwitterionic groups -N^+^(CH_3_)_2_- and -SO_3_^−^. After adding LiCl, the association between groups of opposite charges on the polymer chain was interrupted by Cl^−^ and Li^+^, and a hydrogel structure with uniform group distribution were obtained (Figure 4a). Polyionic hydrogel electrolytes have proven to be ideal materials for regulating ion transport in batteries due to their ion-carrying capacity. In contrast, zwitterionic polymer electrolytes have the advantage of simultaneous regulation of cations and anions. In the interpenetrating network of PVA and zwitterionic salts (ZIS) constructed by Li et al., ZIS provides positively charged pyridine groups and negatively charged sulfonates, and the transport of cation and anion between the electrodes is achieved by selecting the positive and negative groups of the ZIS chain, respectively (Figure 4b). The transport mechanism of separation improves ion transport efficiency [72].

### 4.4. Intelligent Response Characteristic

Traditional hydrogels are insensitive to environmental factors, and their structure and physical and chemical properties are stable for a long time. Smart hydrogels can respond quickly to small changes or stimuli in external environmental conditions (pH, temperature, electricity, magnetic, light) by changing their physical structure or chemical properties [73]. 

Some ionizable weak acid or weak base groups exist in pH-sensitive hydrogel polymers. When the pH condition changes, the corresponding groups will be ionized, generating ion concentration differences between the internal and external environment, and the gel will swell or contract. For instance, polymer chains with weakly acidic groups (such as -COOH) shrink when exposed to low pH, with the increase of pH value, the gel expands with the gradual dissociation of carboxyl groups [74]. The swelling degree reaches an upper limit with complete dissociation of the -COOH. The swelling behavior of the weakly basic group is reversed. Hydrogels with the above two kinds of groups having different properties represent a large pH response at high and low pH values, while the intermediate value is small. It is worth noting that the swelling of hydrogel involves not only a volume change but also a phase change (Figure 5a) [75]. 

The response of hydrogel to temperature is attributed to the change in swelling rate caused by the hydrophilic and hydrophobic transformation. Polyn-isopropylacrylamide (PNIPAM) is one of the most widely studied temperature-sensitive hydrogels. It has a low critical solution temperature (LCST) of about 30 °C. When the temperature is lower than 30 °C, a strong hydrogen bond is formed between hydrophilic groups and water molecules. The hydrogel is in an expansion state by stretching the polymer. When the temperature is higher, the regular hydrogen bond between the molecular chain and water is broken [76]. The hydrophobic isopropyl segment repels water molecules, causing the polymer chain to curl and the gel contracts. Mishra et al. printed a perforated polyacrylamide (PAM) layer on a 3D-printed PNIPAM hydrogel layer to mimic mammalian heat dissipation [77]. When the ambient temperature is above the critical transition temperature (>30 °C), the hydrophilic-hydrophobic conversion occurred in the PNIPAM part. The gel contracted and released water from the open pore structure. This “autonomous sweating” design enhances the thermal stability of the instrument in actual operation.

The magnetic field response can be achieved by adding magnetic particles to the gel, and its response can be explained by magneto-thermal phenomena (Figure 5b) [78]. At zero magnetic field intensity, the magnetic moment in the hydrogel of composite magnetic nanoparticles is randomly distributed. The whole is in a low-energy state. By exposing to a high-energy magnetic field, the magnetic moments become ordered and aligned along the magnetic field. When the magnetic moments return to the ground state, they release heat to the colloid. 

Like magnetically responsive hydrogels, electrosensitive hydrogels can detect the presence of an external field. Still, the mechanisms are distinct since an electric field does not directly drive the bending deformation of a hydrogel. Under the applied electric field, the movable ions in hydrogel will migrate to the corresponding electrode, resulting in an imbalance of ion distribution. Uneven ion distribution results in osmotic pressure difference, and then the water molecules in the hydrogel migrate, bend, and expand state change. Li et al. simulated the electro binding of poly(acrylamide-co-2-acrylamido-2-methylpropanesulfonic acid) [P(AAm-co-AMPS)] hydrogel [79]. Under the applied electric field, movable anions and counter-ions converge to the cathode at the same time. At this time, ion concentration difference is generated between the anode and cathode poles, local dehydration/water absorption occurs in the hydrogel, and the gel bends to the cathode. 

The response behavior of the photoresponsive hydrogel is derived from the photoreactive group. When the hydrogel is irradiated by an external light source, the change of the photosensitive group causes a change in the hydrogel’s expansion state [80]. For example, polymers containing the azo groups or their derivates can undergo the cis-trans reversible structural transformation when light is changed from UV to visible due to the n-π* or π-π* excitations driven by the UV or visible light, respectively. Iwaso et al. designed [c2] daisy chains hydrogels with (aminopropyl diethylene glycol)-4-azobenzene side chain (AmAzoCD) side chains on α-cyclodextrin (αCD) main chains [81]. It realized the state change between the expansion and contraction of the hydrogel system using the isomerization of the azobenzene group by changing the wavelength of the light (Figure 5c).

**Figure 5 polymers-14-04037-f005:**
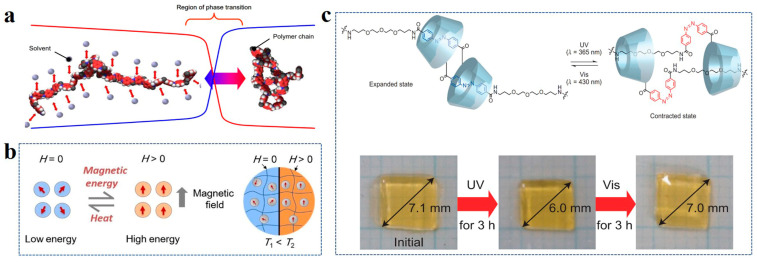
(**a**) Phase transition of polymer chain during swelling. Reprinted with permission from Ref. [75]. (**b**) Magnetothermal phenomena of hydrogels doped with magnetic particles. Reprinted with permission from Ref. [78]. (**c**) Hydrogel volume changes by azo-based light response. Reprinted with permission from Ref. [81].

## 5. Applications

### 5.1. Hydrogel Membrane for Water Treatment 

Water resources account for 71% of the earth in the world. However, the available freshwater resources are much limited. Moreover, due to the progress of modern society, all kinds of industrial wastewater are discharged eventually. Some pollutants, such as heavy metal ions and chemical dyes, inevitably pollute the water body in nature. Therefore, the treatment of water pollution and saltwater desalination is particularly necessary. Many feasible methods, such as coagulation precipitation, biodegradation, ion exchange, membrane separation, adsorption, etc., have been explored [82]. Hydrogel spherical particles have been developed and directly used as adsorbents or designed as membranes to intercept pollutants effectively [83,84]. Adsorption is a method with a relatively wide application range and more development. The hydrogel sorbents can adsorb the pollutants in wastewater. After reaching the adsorption balance, it will be separated from the wastewater to achieve the effect of water purification. Specific surface area from porous hydrogels offers good adsorption capacity, while the ionizable functional groups can adsorb charged substances in water. Generally, adsorption is divided into physical adsorption and chemical adsorption. Physical adsorption is generally van der Waals force and electrostatic attraction, while chemical adsorption is generally chemical bond force or hydrogen bond [85]. The adsorption mechanism of adsorbent materials acting on pollutants is generally not single. For example, when a self-assembled hydrogel network is used for dye removal, electrostatic interaction, hydrogen bonding, pore diffusion, and noncovalent interaction exist simultaneously [86].

#### 5.1.1. Dye Removal

One important factor affecting hydrogel’s adsorption effect on dye is its three-dimensional structure. Yu et al. regulated the content of GO@TAFePc in PVA/GO@TAFePc hydrogels and obtained gel structures with different pore sizes and porosity. The removal rate of methylene blue was up to 98% [87]. Several studies have shown that adding metal oxides can effectively improve the dye removal rate. Barak et al. utilized (N, N-methylenebisacrylamide (NMBA)) to crosslink AMN and TiO_2_. Then, free radical polymerization was initiated by potassium peroxy disulfate to obtain TiO_2_/PAM hydrogel [88]. The dye removal rate was obtained by calculating the peak intensity of the methyl orange UV spectrum. Compared with pure PAM hydrogel, the methyl orange removal rate of TiO_2_/PAM increased by 92.6%. The rate of improvement can be explained in two ways: The addition of TiO_2_ improves the photodegradation ability of dye molecules. Moreover, the swelling property of the gel is decreased to maintain the binding site and binding time between dye and hydrogel.

Most reported hydrogels were used for single dye removal because the adsorbent surface charge and functional interaction cannot act on various dyes. However, dyes in wastewater do not usually exist alone. Therefore, to achieve the simultaneous adsorption of various dyes, multiple specific adsorption sites should be available in hydrogel adsorption materials. Mani used graphene oxide (GO), polyvinyl imine (PEI), and polyvinyl alcohol (PVA) to synthesize the graphene oxide-polyvinyl imide-polyvinyl alcohol hydrogel (GPPH) microsphere with a hybrid network using microwave-assisted reaction [89]. It could adsorb multiple dyes (Figure 6). GO is a good 2D adsorption material due to its large specific surface area and rich oxygen-containing functional groups. However, GO is difficult to be recycled after water treatment. The hybrid microspherical adsorbent overcomes this problem. In addition, the amino and hydroxyl groups contained in PEI and PVA increase the binding sites of dye molecules. The hybrid network strategy can achieve the adsorption of a wide range of dye species simultaneously.

#### 5.1.2. Heavy Metal Ion Removal

Hydrogels prepared from chitosan, acrylamide copolymerization, and natural polymer graft are usually used for heavy metal ion adsorption. As a low-cost raw material for biomass adsorption [90], the main reasons for limiting the adsorption properties of chitosan are poor mechanical properties, a narrow range of application to pH, and insufficient thermodynamic stability. Currently, the proposed solution for low mechanical properties is the crosslinking strategy, which forms stable complexes through covalent/noncovalent crosslinking to improve mechanical properties [91]. 

Like chitosan, polyacrylamide (PAM) is also used as a flocculant. Amino and carbonyl groups in the PAM gel network provide binding sites for metal ions [92]. It is often combined with other components to obtain better ion adsorption properties. Natural anionic polysaccharide carboxymethyl cellulose (CMC) is also a widely developed hydrogel adsorbent with hydroxyl and carboxyl groups [93]. Lu et al. dissolved AM monomer, crosslinking agent, and initiator to prepare the CMC/PAM composite hydrogels. Its adsorption capacities for copper, platinum, and chromium ions were evaluated [94]. The results showed superior adsorption capacities on these ions using separated CMC and PAM-based adsorbents. The transport of ions from the external environment to the gel interior of the tightly structured 3D gel network, as well as the accumulation of the number of CMC and PMA metal ion chelating groups, enhances the ion adsorption capacities. 

Another strategy to improve the adsorption performance of natural polymer hydrogels is to introduce side chains or some functional side groups into the main chain by grafting. For the natural polymer, the limited number of chelation sites can be improved by grafting. For example, lignin is a promising natural renewable hydrogel adsorbent. Jiao et al. synthesized the sulfomethylated lignin-grafted-polyacrylic acid (SL-G-PAA) hydrogel with improved water solubility [95]. The introduction of sulfonymethyl increased the number of active adsorption sites for metal ions without destroying the structure of lignin (Figure 7a). The grafted PAA gel showed improved adsorption efficiency of Co^2+^, Cu^2+^, Ni^2+^, Cd^2+^, and Pb^2+^. 

Most reported adsorbents have no specific selectivity for heavy metal ions, and specific ions cannot be isolated from multi-ion systems. The specific combination of groups and metal ions can be targeted by introducing special functional groups to remove some metal ions. The adsorption properties of chitosan have been widely recognized, and many promising adsorption materials have been developed based on this. The chitosan-graft-polyacrylamide magnetic microspheres (CS-PAM-MCM) developed by Li et al. showed preferential selective adsorption of Hg^2+^ after grafting PAM [96]. In the three-ion system composed of Cu^2+^, Hg^2+^, and Pb^2+^, the adsorption capacity of CS-PAM-MCM for Hg^2+^ is doubled or even increased for Pb^2+^ and Cu^2+^, which is due to the strong affinity of amide groups in PAM for Hg^2+^. 

The adsorption performance of hydrogel is closely related to its structure, and the uniform distribution and compactness of pores will improve the physical adsorption related to the adsorbent structure [97]. Therefore, adjusting the gel network structure can improve the adsorption efficiency. A semi-interpenetrating network was prepared by introducing linear polyvinyl alcohol (PVA) into the copolymer crosslinking of sodium alginate-grafted-sodium polyacrylate (SA-g-PNaA) [98]. The formation of this structure was confirmed by Fourier Transform Infrared Spectroscopy (FTIR), which showed the movement of PVA characteristic absorption peak, indicating hydrogen bond interaction between the polymer network and PVA. Compared with SA-g-PNaA with a few pores and gaps, SEM showed large pores and a loose surface of SA-g-PNaA/PVA (Figure 7b). To some extent, the pore size and surface porosity increase with the introduction of PVA. As the semi-IPN material with PVA added adjusts the pore structure of the matrix, the increase of pores amplifies the specific surface area of physical adsorption. The homogenization of the network also reduces the spatial transport resistance of ions in the matrix, achieving high ion transport efficiency. 

#### 5.1.3. Water Desalination

Desalination of seawater by solar energy is a hot topic of sustainable concept. The core element is the photothermal conversion materials. In solar-driven evaporation systems, the pore structure of hydrogel-based photothermal conversion materials could be taken for a rich channel for water transport for steam transport. The hydrophilic groups in the hydrophilic polymer network are beneficial in reducing the evaporation enthalpy of water molecules, accelerating water transport, and improving the solar energy conversion efficiency. The capture of sunlight and water transport are the core aspects of a solar thermal converter. Guo et al. introduced Fe-MOF into konjac Glucomannan (KGM) and polyvinyl alcohol (PVA) hydrogel networks and produced mixed hydrogel evaporator (HHEs) by in situ polymerization. The photothermal nanoparticle Fe-MOF solar absorbent converts solar energy into heat and generates purified steam in situ [99]. Two-dimensional materials with excellent light absorption and photothermal conversion ability have also been developed for hydrogel evaporators. Li et al. used directional freezing technology to synthesize reduced graphene oxide (A-RGO) hydrogels. The hydrogel frame was coated with a hydrophilic Ti_3_C_2_Tx MXene sheet to construct a hybrid hydrogel with regular vertical channels (Figure 7c) [100]. Vertically arranged pores facilitate the transport of water molecules to the surface, and the preservation of abundant oxygen-containing functional groups is conducive to reducing water enthalpy of evaporation, improving the rate of solar steam generation. The excellent mechanical properties of 2D material effectively prevent the shrinkage of gel polymers under high power irradiation.

**Figure 7 polymers-14-04037-f007:**
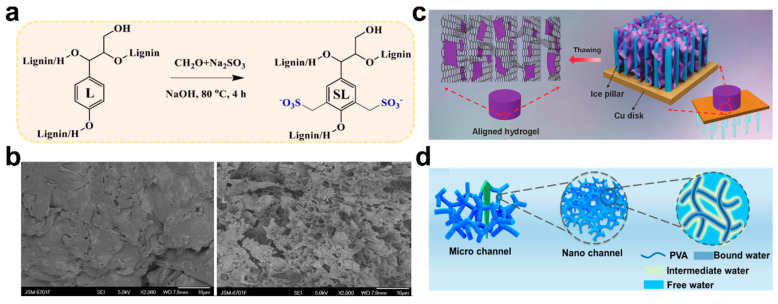
(**a**) Sulfonomethyl treatment of lignin. Reprinted with permission from Ref. [95]. (**b**) SEM micrographs of SA-g-PNaA (left) and SA-g-PNaA/PVA (right). Reprinted with permission from Ref. [98]. (**c**) Schematic diagram of Ti_3_C_2_Tx MXene coated vertical channel reduced graphene oxide (A-RGO) hydrogel. Reprinted with permission from Ref. [100]. (**d**) Exhibition of hierarchical channel structure in the hydrogel. Reprinted with permission from Ref. [101].

The water purification efficiency of solar evaporators is affected by energy loss. Firstly, the evaporator receives a limited amount of radiant energy from the sun. Secondly, the thermal radiation and convection loss dissipated in nature will occur after the evaporator’s photothermal surface absorbs energy. To solve such problems, Yang et al. designed a membrane covering the surface of the evaporator to reduce energy dissipation [101]. It is equipped with a transparent hierarchical porous structure (Figure 7d). The highly transparent hydrogel does not block the heating layer from receiving solar radiation. The polyvinyl alcohol (PVA) micron scale network was prepared by introducing glycerol and undergoing a freeze–thaw cycle. This structure ensures the water content of the gel membrane and contributes to water evaporation. The nanoscale structure is formed using polyethylene glycol (PEG) as a pore-forming agent on the micron’s channel wall. This layer provides superior hydration. The convection loss is eliminated by the membrane blocking the contact between the air and the evaporator. The thermal radiation generated by the photothermal surface is recaptured by the gel membrane and used again to produce water vapor. After covering the gel film, the evaporation efficiency reaches 95%, and the heat radiation and convective loss decrease from 6.6% to 0.39%. 

### 5.2. Salinity Gradient Energy Conversion

Since river water and seawater are two fluid systems with different salinity, osmosis happens when they converge. The energy that comes from nature is enormous. If this energy could be collected, it could provide a new solution to the energy crisis. Nowadays, salt concentration differential energy extraction methods, including pressure retarded osmosis (PRO), vapor pressure deficit (VPD), and reverse electrodialysis (RED), could provide energy [102,103]. RED, the most commercialized technology, utilizes selective and permeable membranes to capture permeable energy [104]. The performance of the membrane largely determines the level of energy conversion efficiency [105]. Hydrogel materials are widely used as ion-selective membranes for salinity gradient energy conversion [106].

The use of salinity gradient to generate electricity is inspired by nature. Electric eels can generate 300–800 V output voltage for defense and to capture prey [107,108]. Its excellent discharge ability comes from the ion selectivity of the cell membrane at the back of its body [109]. There are differences in salinities between the inner and outer compartments of the cell. Nerve stimulation revealed the opening of specific ion channels at the front and back membranes. The selective transport of ions produced a transmembrane potential. The bionic ion-selective membranes were designed by mimicking the electric eel discharge mechanism. Schroeder et al. used different polyacrylamide (PAM) hydrogel components to replace the components of electric eel discharge cells [110]. High-salinity gels alternate with low-salinity gels in the cell’s internal and external environment. Referring to the front and back membranes of electric eel discharge cells, high and low salinity gel chambers were separated by gels with anion and cation selectivity. The hydrogels of four different ingredients form the basic units of the concentration cell. A basic unit could produce a voltage of 130–185 mV at an open circuit, equivalent to a single electric eel cell. Moreover, the potential difference obtained by multiple gel units in serious is also linearly increased, identical to the effect produced by multiple electric eel cells superimposed on each other (Figure 8a).

Recently, ion-selective membranes based on hydrogel materials have been extensively researched. The Cu^2+^ crosslinked sodium alginate (SA) (Cu^2+^-SA) hydrogel membrane developed by Chen et al. achieved the power density conversion of 4.55 W m^−2^ under the simulated environment of sea-river salinity difference [111]. Hydrogels exhibit excellent ion transport capacity due to their various space charges and three-dimensional interconnected channels. Wen et al. conducted a PNP simulation of the synthesized 2-hydroxyethyl methacrylate (HEMAP) hydrogel membrane. Under the same test conditions, the electrical conductivity and power output of the HEMAP channel interconnection model are higher than those of the separated channel model (Figure 8b) [112]. It illustrates the ability of three-dimensional interconnected channels to improve transmission.

**Figure 8 polymers-14-04037-f008:**
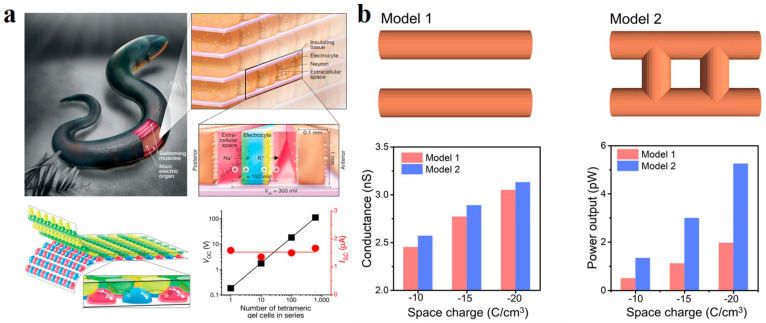
(**a**) A hydrogel generator that mimics an electric eel cell. Reprinted with permission from Ref. [110]. (**b**) The conductivity and power output of the two channels were compared under PNP simulation. Reprinted with permission from Ref. [112].

Hydrogel is a membrane-based power generation material with its unique advantages: abundant transport channels for moving ions. However, the pore size varies from micron to nanometer, most of which are micron, limiting ion transport at the nanometer scale. In addition, low mechanical strength is not conducive to practical application. Therefore, composite materials based on hydrogel have been developed. Natural fibers have neatly arranged nanochannels that carry a negative charge on the surface, facilitating ion transport, and the raw material is economically available. Chen et al. isolated part of lignin and cellulose from a large area of balsa wood sheet and prepared the hydrogel hybrid film by in situ polymerization of polyvinyl alcohol (PVA)/acrylic acid (AA) in the channel [113]. Cellulose and hydrogels are closely bonded by hydrogen bonds, and crosslinked hydrogels in wood channels provide additional nanofluid channels and abundant charge and act as fillers to block large pore sizes.

Human muscle is indispensable in signal transmission with its ordered hydrogel fiber structure. Inspired by this, utilizing natural wood and polyacrylamide (PAM) polymer, a high strength, anisotropic and ionic conductive hydrogel was synthesized by Kong et al. [114]. The orientated cellulose nanofibers were obtained by removing lignin. Ammonium persulfate (APS) as initiator and N, N′-methylene bisacrylamide (MBA) as crosslinking agent initiated the free radical polymerization of acrylamide (AM) in the channel and bonded with CNF by a hydrogen bond (Figure 9a). CNFs from natural wood are strongly crosslinked with PAM. The tensile strength of the obtained wood hydrogel is up to 36 MPa. It is worth noting that, as a natural nanofluid conduit, it presents the ion transport phenomenon similar to a biological system such as muscle. With the negative charge of an abundant hydroxyl group on the surface providing ion selectivity, conductivity shifts were confirmed both at low electrolyte concentrations. Wood hydrogels may be widely used in energy conversion.

Ion channels in living organisms usually have asymmetric structures, which can be switched on and off to achieve controllable ion transport [115]. Bionic ion channels with asymmetric structures also show excellent energy conversion performance [116]. The asymmetric membrane with an ion diode effect can be used as an energy conversion generator component to assist one-way ion transport in the channel [117]. Therefore, the energy conversion efficiency is higher than that of a symmetric membrane. Hybridization of two different porous membranes is one of the methods to prepare asymmetric membranes, but the pore mismatch between them reduces the advantages of asymmetric membranes [118]. Zhang et al. designed an asymmetric membrane by hybridizing Agarose/Polyacrylazulfonate (PSS) and Arlene nanofiber (ANF) membrane (Figure 9b) [119]. In addition to providing strong support, ANF has abundant nanoscale channels. The hydrogel layer is super hydrophilic, and the special 3D channel structure can achieve a high matching degree with the channel of the ANF layer. As an ion diffusion accelerator, the interfacial transmission efficiency is greatly improved. 

However, for hybrid heterogeneous membranes, interfacial resistance limits the ion transport at the interfaces with restricted energy conversion efficiency improvement. This is an inherent drawback of hybrid asymmetric structures [120,121]. It is a major problem for hybrid membranes to avoid interfacial resistance while preserving the function of heterogeneous membranes to promote one-way ion transport. To solve this problem, our group prepared an “integrated” gradient polyelectrolyte hydrogel membrane as an efficient energy conversion generator [122]. The preparation strategy of hydrogel membrane is: when the solution of high and low concentrations contact each other, the reaction rate is different due to the concentration difference, producing a higher complex density on the high concentration side. After the reaction is completed, the density of the structure decreases by a gradient from a high to a low concentration (Figure 9c). In contrast, the negative charge density distribution follows the opposite gradient. The variation of the aperture size was observed by a confocal laser scanning microscope (CLSM), and the gradient distribution of Rhodamine 6G proves the gradient change of charge. The effect of gradient hydrogel membrane for ion transport is obvious. The highest output power density reaches 7.87 W m^−2^ between real river water and the seawater environment. Another outstanding feature of it is its resistance to expansion. Hydrogels in the water environment will have high swelling due to osmotic pressure difference, resulting in the corresponding reduction of mechanical properties with reduced membrane life. Moreover, if a section or part of the network structure breaks, the imbalance between permeability and elastic energy will also lead to swelling. The concentration of ions in the ocean is not high enough to destroy the membrane. It will also bring electrostatic shielding to reduce osmotic pressure in the membrane and avoid excessive swelling, making it stable and durable in practical use. 

**Figure 9 polymers-14-04037-f009:**
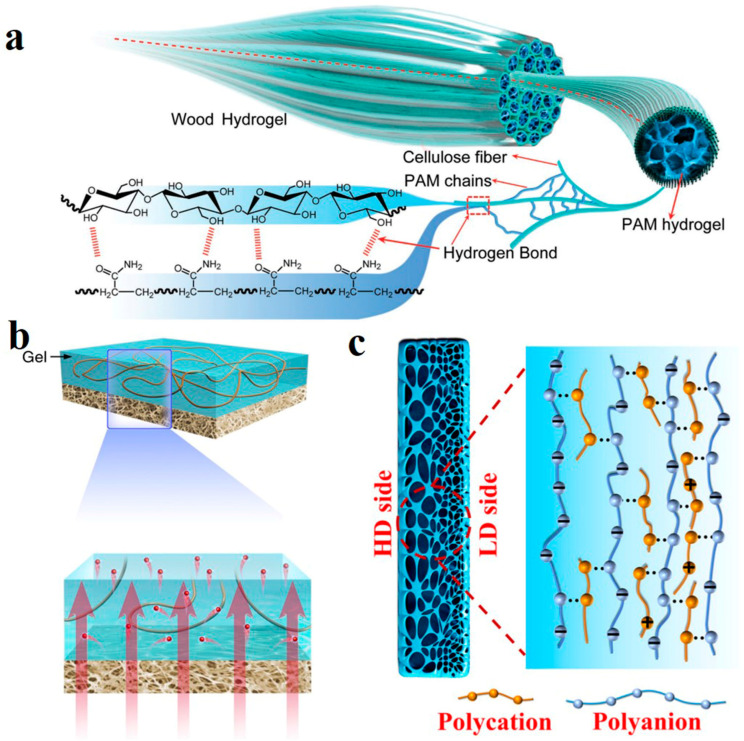
(**a**) Hydrogen bonding between CNF and PAM. Reprinted with permission from Ref. [114]. (**b**) Simulation of hydrogel/ANF heterogeneous membrane. Reprinted with permission from Ref. [119]. (**c**) Schematic diagram of gradient structure presented by hydrogel membrane. Reprinted with permission from Ref. [122].

### 5.3. Energy Storage

An electrolyte is one of the core components of typical energy storage devices such as batteries and supercapacitors [123]. In addition to meeting the necessary conditions of high ionic conductivity, high chemical stability and safety should be considered in normal working conditions. Although solid and liquid electrolytes are relatively mature and widely used, there are still some inherent defects. For example, liquid electrolytes are flammable and explosive. Solid electrolytes have relatively high internal resistance [124]. Hydrogels are regarded as high-potential electrolyte materials transporting ions and as separators between the poles [125]. The main parameters to measure the performance of electrolytes are ionic conductivity—to achieve efficient ion transport; mechanical strength—good mechanical properties; and flexibility—repeated deformation ability. The high ionic conduction efficiency of hydrogel electrolytes results from its 3D interconnectional unique structure and movable path for the ions provided by high water content.

However, hydrogel electrolytes used in extreme temperatures are greatly limited by a large amount of water in the hydrogel body. For example, when the temperature drops below 0 °C, water molecules in the hydrogel bind with strong hydrogen bonds, causing the gel to freeze. Improving the freezing resistance of electrolytes starts with lowering the freezing temperature. Hydrogen bond interactions between water molecules can be weakened by adding auxiliary materials. Huang et al. added low-concentration zinc ion salt Zn(ClO_4_)_2_ into carboxymethyl chitosan (CMCS) -polyacrylamide (PAM) hydrogel, ClO_4_^−^ anion forms ternary interactions with PAM chains and water molecules (Figure 10a), which thus reduces the bonding between water molecules and lowers the freezing point [126]. Polyols have also been shown to be effective in lowering freezing points. Jiang et al. introduced glycerol molecules into hydrogel components which are difficult to freeze at −50 °C [127]. This is because glycerin breaks the bonds between water molecules and reduces the number of free water molecules (Figure 10b). Some organic solvents can also do this. Lu et al. combined dimethyl sulfoxide with polyvinyl alcohol (PVA), a good organic solvent that dissolves in any proportion of water [128]. The interaction between the two will weaken the interaction between water molecules, lowering the freezing point of hydrogel to −50 °C. They also considered that when the ambient temperature is too high, hydrogel loss of water will lead to structural stability damage. Therefore, montmorillonite (MMT), which can improve the thermal performance of the polymer, is compounded into the gel, resulting in an electrolyte material with good thermal stability. Its cold resistance and thermal stability broaden its application scope in extreme conditions.

Since flexible devices will repeatedly deform during the application, hydrogel electrolytes should have strong tensile and compression ability under certain stress. Huang et al. used vinyl hybrid silica nanoparticles (VSNPs) crosslinking polyacrylamide (PAM) to facilitate a dynamic crosslinking network [129]. VSNPs act as a stress buffer to dissipate energy when stress is applied. Unlike conventional polymer chains that can only break under high stress, VSNPs-PAM can realize dynamic fracture recombination of polymer chains to disperse stress, thus achieving overstretch and compressibility (Figure 10c). Water in hydrogel electrolyte serves as a continuous medium to guarantee high flux ion migration. Still, this structure with high water content limits the enhancement of strength. Liu et al. redesigned ion migration paths in hydrogels from the perspective of solvents [130]. Zwitterionic ionic polymer monomer 2-(Methacryloyloxy) Ethyl Dimethyl -(3-sulfopropyl) Nitrogenous material (DMPS) is Crosslinked by highly hydrophilic vinyl silica Microparticles (VSMPs), lithium BIS (trifluoromethanesulfonyl) imide (LiTFSI) was added to obtain a hydrogel electrolyte with interfacial wetting water. The crosslinked zwitterionic polymer framework provides abundant interfacial wetting channels for anion and cation migration. The three-dimensional structure provided by the water block in the traditional hydrogel is replaced by a two-dimensional conduction pathway, which overturns the traditional hydrogel pathway. The combined wetting water at the polymer network interfaces and the water chemically bonded to the frame, together with the water dissolving LiTFSI promote ion transport. Linear sweep voltammetry (LSV) showed that the polymer hydrogel had a large chemical window. Due to the significant reduction of water content, the strength and elongation are significantly improved (Figure 10d), with an ultimate tensile strength of 420 kPa and elongation of more than 6000%. This work provides a new starting point for applying hydrogel electrolytes in flexible energy storage devices. 

### 5.4. Sensors 

A sensing device quickly converts external information into electrical signals or other forms to meet people’s subsequent collection and processing of information [131]. It is an indirect device to obtain external information, widely used in environmental monitoring, medical diagnosis, resource exploration, and other fields. The main superiority of hydrogel flexible sensing materials lies in the similarity of biological tissue structure and mechanical properties [132]. Hydrogel-based sensors are highly compatible with the human body and can be used to monitor human movement and health. It transforms the physiological information of the human body into electrical information with good biocompatibility and high sensitivity. 

Security is also a core concern that should be paid attention to. A barrier is often used as a protective layer for hydrogel sensors on human skin to ensure safety, but this additional layer may make the sensor less sensitive. Ma et al. prepared a two-layer hydrogel sensor to achieve skin protection and rapid response [133]. The skin protection layer is made of PVA mixed with silicone oil, and the conductive layer is doped with polymer polyaniline with good electrical conductivity (Figure 11a). The interface blocks the passage of polyaniline, effectively protecting the skin. Compared with the conventional single-layer hydrogel sensor, which requires an additional isolation layer, the accuracy was improved by 32.1%. 

Most sensors are based on wired communication and are relatively stable, while wireless sensors offer greater flexibility and avoid wire distortion or even breakage due to motion deformation. Xiong et al. designed a polyglycol diglycidyl ether (PEGDE) crosslinked DNA hydrogel that enables wireless communication based on the interaction between DNA strands and deoxyribonucase (DNase). When the gel matrix is exposed to the DNase environment produced by the bacteria, the DNA crosslinking network breaks, and the gel matrix structure is destroyed (Figure 11b) [134]. This change is sensed by embedded finger-like electrodes and transmitted via near-field communication to a smart device that accurately reflects the number of bacteria for wound management (Figure 11c).

Liang et al. demonstrated the oxygen-sensitive hydrogel principle in their synthesized polyacrylamide-chitosan (PAM-CS) hydrogel sensors [135]. When the sensor is exposed to a certain oxygen concentration, the oxygen reacts to gain electrons at the cathode, and the silver electrode loses electrons at the anode (Figure 12a). The whole system constitutes an electrolytic cell. The response is measured by the system current. The response intensity increases linearly with the ascending of oxygen concentration due to the enhancement of current intensity caused by the increased reaction intensity (Figure 12b). In addition, hydrogel sensors will inevitably face the problem of poor durability caused by the evaporation of water molecules when exposed to the environment. Covering the hydrogel surface with a thin film is an effective method to prevent evaporation. However, it is important to note that the sensor’s sensitivity must be guaranteed after covering the film. Ye et al. prepared a porous ecoflex membrane with salt as a sacrificial layer template. They assembled an integrated oxygen sensor with the UV-induced covalent bond between the ecoflex membrane and carrageenan/polyacrylamide (PAM) hydrogel (Figure 12c) [136]. The open membrane structure ensures that the sensor receives oxygen. In the water retention test, the water retention rate of the uncoated device is less than 50%, while the water retention rate of the coated device could reach more than 70%. In conclusion, applications related to ion transport in hydrogels are summarized in Table 1.

## 6. Conclusions

As a soft and wet material that can retain water and deform, hydrogels have continuously attracted much attention in recent years. The special 3D crosslinking structure and the amazing water carry capacity of up to 1000 times dry weight make it a material with great application prospects and potential. This paper reviews the latest research on hydrogel-based materials in their origin, construction, structural characteristics, and mass transport applications. In this review, we discussed hydrogel materials from different sources. Natural hydrogels have inherent good biocompatibility, while the development of synthetic hydrogels is more targeted, and hybridization is a complementary strategy. In addition, the basic structure mechanism of hydrogel, including crosslinking mechanism and space structure construction, was summarized. In the structural design of hydrogels, chemically crosslinked hydrogels show stronger strength than physically crosslinked networks due to the difference in strength caused by chemical bonding force. The difference in strength also lies in the network construction. The mechanical properties of double polymer networks are far better than that of single networks. Pore structure and size distribution can be customized by preparation conditions. Water content is closely related to mechanical properties. In addition, hydrogels also have ion-carrying properties, showing excellent mass transport performance. For example, hydrogels exert excellent physical adsorption capacity due to their 3D spatial network. Specific heavy metal ions or dye ions can be selectively targeted by functional group modification of the polymer substrates. With high transparency and good ion permeability, water purification can be achieved by hydrogels as the emerging adsorbent and water purification materials. The ionic conductivity property of hydrogel provides a broad reaction platform for electrochemical applications. With high flexibility and a high degree of mechanical matching with human movement, it can sensitively capture the tiny changes in the human body state. In the field of salinity gradient energy conversion, enhanced strength and biomimetic nanofluid channel membranes have been successfully obtained. By constructing the asymmetric structures in the hydrogel membranes, one-way ion transport can be effectively promoted, resulting in considerably improved energy conversion efficiency. As one of the components of reverse electrodialysis power generation devices, hydrogel membrane is a great subversion to traditional solid membrane materials and offers new possibilities for clean, renewable energy capture.

Despite excellent flexibility and ion transport in a 3D network by hydrogel materials, the mechanical strength of traditional rigid devices is still unmatched by the hydrogel. For instance, ionic conductive hydrogels with wide electrochemical windows are ideal frame materials for energy storage and conversion devices. It is still one of the most promising materials for developing highly demanded modern, flexible, intelligent, and portable electronic devices. However, there is still a certain gap for practical applications. The limitation of insufficient strength and the inevitable water loss are the major obstacles in practical applications, resulting in inadequate energy conversion efficiency or limited component service life during the energy storage and conversion cycle. It is impossible to transport clean water stably and efficiently. Hydrogels with these limitations have not yet reached commercial standards. The development of hydrogel components still needs reasonable structure and performance adjustment. The selection of monomer, polymer types, and crosslinking mode are several strategies for adjustment. At the same time, the preparation method or composite high-strength materials are involved in achieving the effect of strength enhancement.

## Figures and Tables

**Figure 2 polymers-14-04037-f002:**
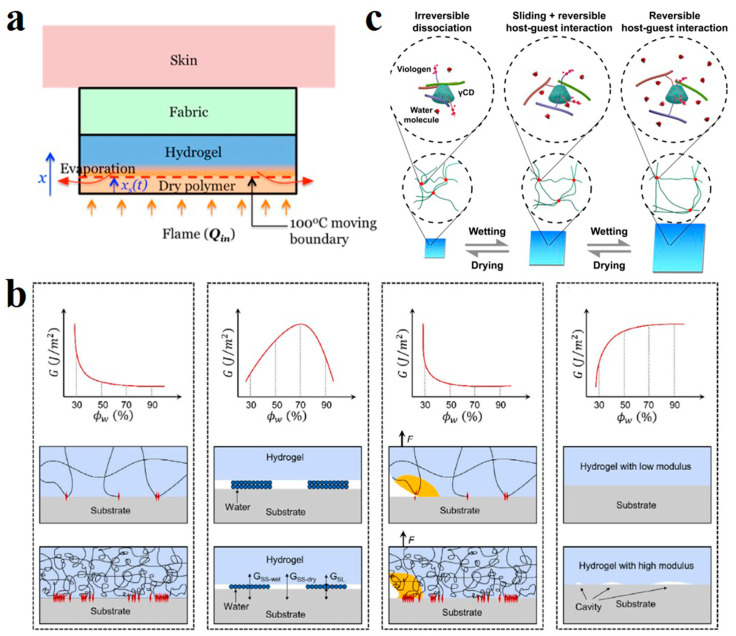
(**a**) Heat dissipation mechanism of polyacrylamide-alginate hybrid hydrogels layers when exposed to open fire. Reprinted with permission from Ref. [56]. (**b**) Four factors are involved in the relationship between water content and the adhesion energy of hydrogel. Reprinted with permission from Ref. [57]. (**c**) Polymer chain slip or host–guest interaction in hydrogels with different water contents. Reprinted with permission from Ref. [58].

**Figure 3 polymers-14-04037-f003:**
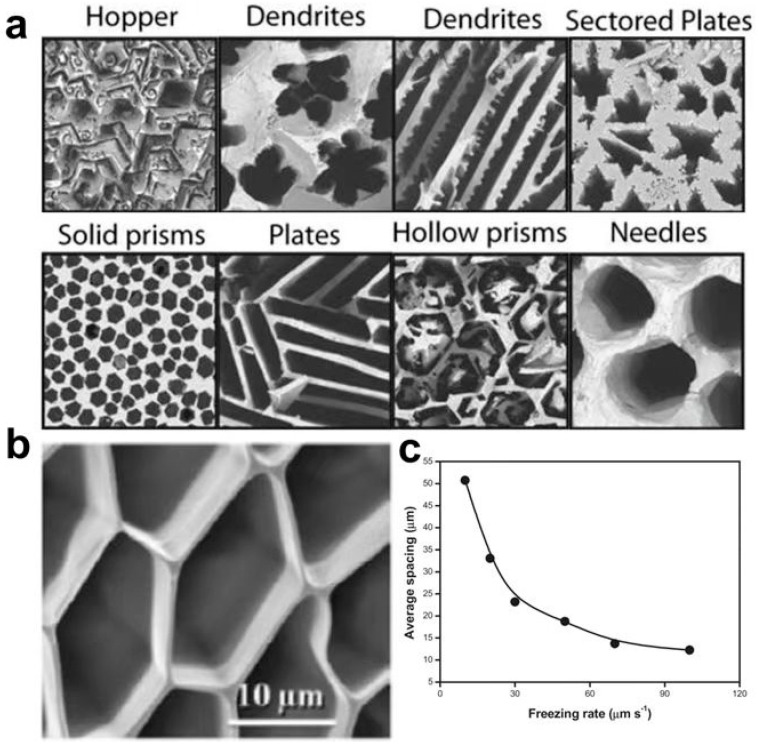
(**a**) Typical hydrogel pore morphology obtained by controlling ice crystal growth. Reprinted with permission from Ref. [63]. (**b**) SEM cross-section of regular PVA hydrogels obtained by directional freezing technique. (**c**) Relationship between freezing rate and pore size. Reprinted with permission from Ref. [64].

**Figure 4 polymers-14-04037-f004:**
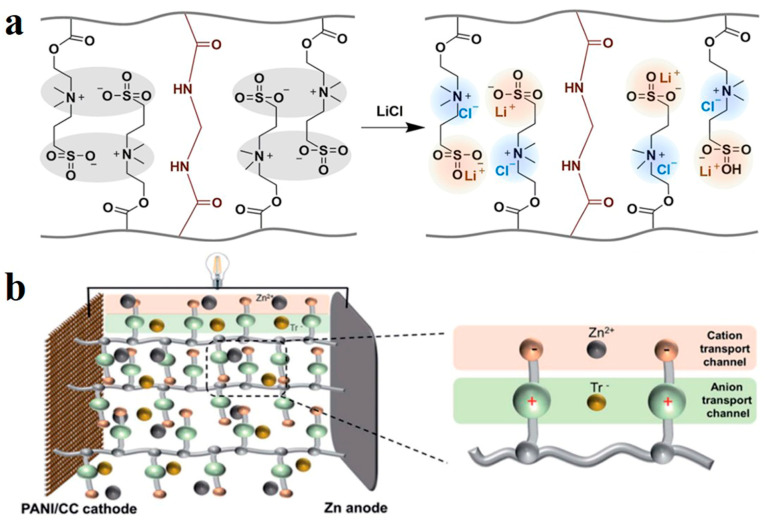
(**a**) Mechanism of LiCl blocking polymer chain association. Reprinted with permission from Ref. [71]. (**b**) Simulation diagram of independent anion and cation transport channel. Reprinted with permission from Ref. [72].

**Figure 6 polymers-14-04037-f006:**
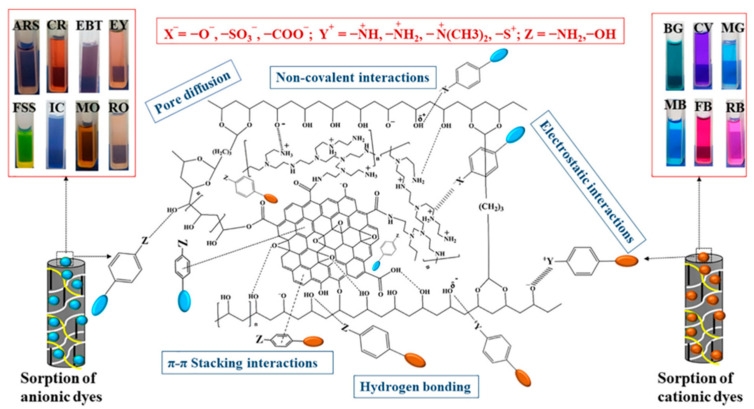
Multi-site dye adsorption mechanism diagram of GPPH microspheres. Reprinted with permission from Ref. [89].

**Figure 10 polymers-14-04037-f010:**
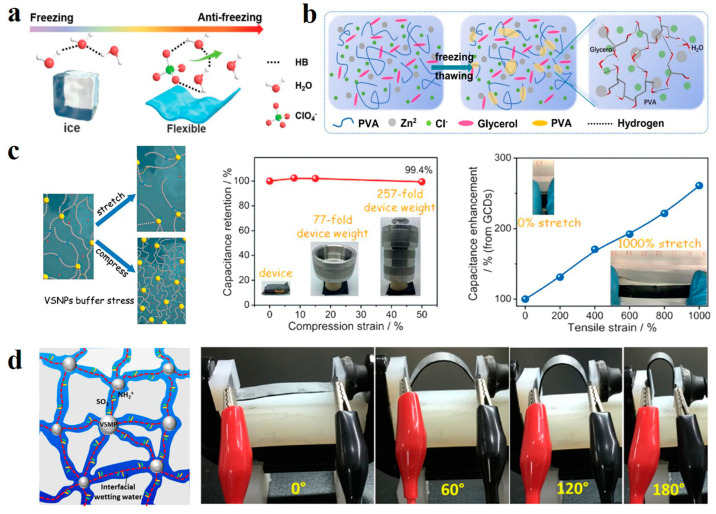
(**a**) Antifreezing schematic of ClO_4_^−^ ternary interaction with PAM chains and water molecules. Reprinted with permission from Ref. [126]. (**b**) Mechanism of action of glycerol molecule in antifreeze hydrogel. Reprinted with permission from Ref. [127]. (**c**) Overstretch and compressibility achieved in dynamic crosslinked networks. Reprinted with permission from Ref. [129]. (**d**) High deformability of interfacial wetting water hydrogel electrolytes. Reprinted with permission from Ref. [130].

**Figure 11 polymers-14-04037-f011:**
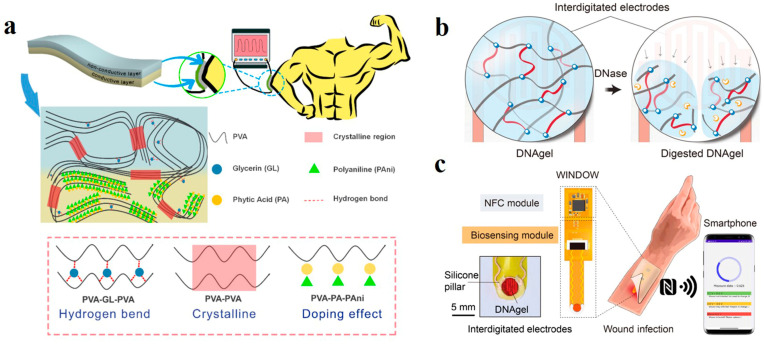
(**a**) Schematic diagram of double hydrogel protecting skin. Reprinted with permission from Ref. [133]. (**b**) The breaking mechanism of DNA hydrogel. (**c**) The principle of wireless communication. Reprinted with permission from Ref. [134].

**Figure 12 polymers-14-04037-f012:**
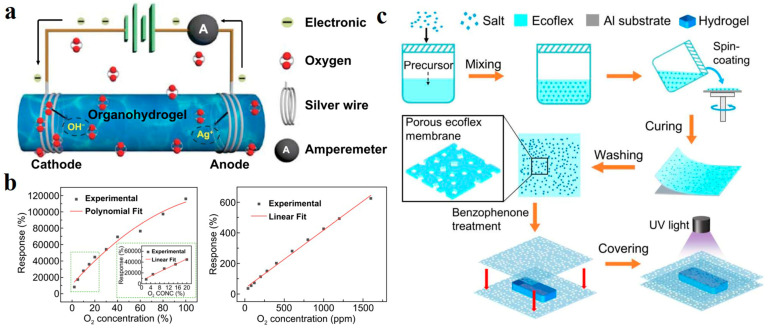
(**a**) The principle of hydrogel sensor. (**b**) The linear relationship between response intensity and oxygen concentration. Reprinted with permission from Ref. [135]. (**c**) Schematic diagram of moisture resistant oxygen sensor assembled on hydrogel surface using ecoflex porous membrane prepared by sacrificial template. Reprinted with permission from Ref. [136].

**Table 1 polymers-14-04037-t001:** Applications related to ion transport in hydrogels.

**Ion Transport Related Applications**		
Hydrogel membrane for water treatment	Dye removal Heavy metal ion removal Water desalination	Refs.: [82,83,84,85,86,87,88,89,90,91,92,93,94,95,96,97,98,99,100,101]
Salinity gradient energy conversion	Ion-selective membranes	Refs.: [102,103,104,105,106,107,108,109,110,111,112,113,114,115,116,117,118,119,120,121,122]
Energy storage	Anti-freezing electrolyte diaphragm Highly flexible deformable electrolyte diaphragm	Refs.: [123,124,125,126,127,128,129,130]
Sensors	Ambient sensors Human body sensors	Refs.: [131,132,133,134,135,136]

## Data Availability

Not applicable.

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
