# Peer review of "Construction and Ion Transport-Related Applications of the Hydrogel-Based Membrane with 3D Nanochannels"

_polymers, 2022, doi:10.3390/polym14194037_

Round 1

Reviewer 1 Report

In this manuscript, the authors wrote a review article about Hydrogel, Hydrogel is a type of crosslinked three-dimensional polymer network structure gel. They focus on ion transport-related applications based on hydrogel materials. They briefly elaborated on the origin and source of hydrogel materials and summarized the crosslinking mechanisms involved in matrix network construction and the different spatial network structures. They summarized the prospects and challenges of hydrogels. The interpretations of the published results on hydrogel are well discussed and summarized. In addition, the quantity and quality of the figures are appropriate. We believe that this review subject is good and important to provide essential data and meaningful references data for hydrogel-based membranes with their applications such as water treatment, Salinity gradient energy conversion, Energy storage, and sensors,  

Summary: I recommend publishing this review after considering my comments on the attached file.

Author Response

Thanks for your comment on the attached file. We have added “the role of the free volume with water uptake in the polymers” and corresponding references in the revised manuscript. (section 4.1). “With the increase in water absorption, the free volume of water in the polymer chain increases [59]. The increased free body water acts as a plasticizer, making the polymer molecular chains slip so that the molecular chain movement is enhanced [60,61].”

Reviewer 2 Report

Dear Authors

The presented review is very interesting for the readers and addressed an essential topic. The review focus on ion transport-related applications based on hydrogel materials. We briefly elaborated on the origin and source of hydrogel materials and summarized the crosslinking mechanisms involved in matrix network construction and the different spatial network structures. Then, hydrogel structure and the remarkable performance features such as microporous, ion carrying, water content, and responsiveness for pH, light, temperature, electricity, and magnetic field have been discussed. Moreover, we focus on the application of hydrogels in water purification, energy storage, sensing, and salinity gradient energy conversion. Finally, the prospects and challenges of hydrogels are summarized.

The review at present fo is acceptable and has a general overview.

However, missing points have to be addressed such as:

1-The effect of different physical forms of hydrogel on its characters,

2- The crosslinking of the hydrogel using different radiation techniques,

3- The bioapplications of the hydrogels especially in tissue engineering applications, drug delivery applications, and so on.

In conclusion, further modification needs to have a wider overview review.

Author Response

Reviewer 2

The presented review is very interesting for the readers and addressed an essential topic. The review focus on ion transport-related applications based on hydrogel materials. We briefly elaborated on the origin and source of hydrogel materials and summarized the crosslinking mechanisms involved in matrix network construction and the different spatial network structures. Then, hydrogel structure and the remarkable performance features such as microporous, ion carrying, water content, and responsiveness for pH, light, temperature, electricity, and magnetic field have been discussed. Moreover, we focus on the application of hydrogels in water purification, energy storage, sensing, and salinity gradient energy conversion. Finally, the prospects and challenges of hydrogels are summarized.

The review at present is acceptable and has a general overview.

However, missing points have to be addressed such as:

Comment 1: The effect of different physical forms of hydrogel on its characters,

Response: Thanks for your comments. We have added these in the revised manuscript. (section 3.2).

“In comparison to the simple single polymer network structure, the dual network is improved in both mechanical strength and toughness due to the presence of the second network. At present, it is applied in biomedicine, biomimetic machinery, and other fields, which have a higher demand for strength and toughness.

“The kind of physical structure formed is characterized by the unique forcing effect of the topological structure, which can form stable binding between two or more polymers with very different properties or different functions, thus achieving special coordination in properties and structures, and greatly improving mechanical strength.”

Comment 2: The crosslinking of the hydrogel using different radiation techniques,

Response: Thanks for your comments. We have added these in the revised manuscript. (section 3.1).

“High-energy electron beams and gamma rays are the main radiation conditions to in-duce the cross-linking of the polymer network. Under irradiation conditions, the water particles in the gel solution produce living groups, which react with the polymer by hydrogen extraction to form polymer free radicals, and the free radicals are cross-linked to form hydrogel network polymers.”

Comment 3: The bioapplications of the hydrogels especially in tissue engineering applications, drug delivery applications, and so on.

Response: Thanks for your comments. Due to high hydrophilicity, high permeability, good biocompatibility and low friction coefficient, hydrogels have broad application prospects in biomedicine, tissue engineering and other fields. For example, hydrogels are equipped with channels to deliver drug molecules, which, when combined with responses in different physiological environments, can achieve targeted drug release in the human body. In addition, the soft, moist surface and tissue affinity reduce the irritation to human tissues, which has a unique advantage in the field of tissue repair. We have thought carefully about your suggestion, but since the original intention of this paper is to focus on ion transport applications related to hydrogel electrochemistry rather than biomedicine. Thus, we did not add this part.

In conclusion, further modification needs to have a wider overview review.

Reviewer 3 Report

Manuscript title: Construction and ion transport related applications of hydrogel-based membrane with 3D nanochannels

Type of manuscript: Review

Manuscript ID: polymers-1911002

Review comments: The review is interesting and valuable to the readers. It describes about the application of hydrogels in water purification, energy storage, sensing, and salinity gradient energy conversion, the prospects and challenges of hydrogels and study of Construction and ion transport related applications of hydrogel-based membrane with 3D nanochannels. But it needs the following corrections before accepting its final publication.

1.    In abstract, it needs to be re-written in one particular form. It is mixed of past and present tense. Mixing is a wrong method.

2.    In Keywords it needs particular keyword name. It is absent there.

3.    The introduction part is insufficient. It needs to be added more references to be improved significantly the manuscript.

4.    In Figure 3a needs to be high magnification.

5.    In Figure 6a should be separate.

6.     Need one table for all applications and the corresponding refs.

7.     Use the reproduced figure from the ref with the permission indicating the ref number.

8.    The English is need to edit moderately.

Author Response

Review 3

Review comments: The review is interesting and valuable to the readers. It describes about the application of hydrogels in water purification, energy storage, sensing, and salinity gradient energy conversion, the prospects and challenges of hydrogels and study of Construction and ion transport related applications of hydrogel-based membrane with 3D nanochannels. But it needs the following corrections before accepting its final publication.

Comment 1. In abstract, it needs to be re-written in one particular form. It is mixed of past and present tense. Mixing is a wrong method.

Response: Thanks for your comments. We have unified tenses in abstracts, the present tense is used in the revised manuscript.

Comment 2. In Keywords it needs particular keyword name. It is absent there.

Response: Thanks for your comments. We have added the keywords to the corresponding locations in the revised manuscript. “hydrogels; 3D structure; ion channels”

Comment 3. The introduction part is insufficient. It needs to be added more references to be improved significantly the manuscript.

Response: Thanks for your comments. We reviewed the literature and added more references to the introduction part in the revised manuscript.

“Ref:[5], [9], [10], [11], [12], [14], [15], [22], [25], [26], [27], [28].”

Comment 4. In Figure 3a needs to be high magnification.

Response: Thanks for your comments. We have magnified Figure 3a appropriately in the revised manuscript.

Comment 5. In Figure 6a should be separate.

Response: Thanks for your comments. We have separated Figure 6a in the revised manuscript.

Comment 6.  Need one table for all applications and the corresponding refs.

Response: Thanks for your comments. We have added one table for all applications and the corresponding refs in the revised manuscript.

Comment 7.  Use the reproduced figure from the ref with the permission indicating the ref number.

Response: Thanks for your comments. We have added this to all referenced captions in the revised manuscript.

Table 1. Applications related to ion transport in hydrogels.

Comment 8.  The English is need to edit moderately.

Response: Thank you for your comment, we have revised the language in the revised manuscript.

Round 2

Reviewer 3 Report

The manuscript has been improved after careful revision. I think it is accepted in the current form.